

# High variability within pet foods prevents the identification of native species in pet cats' diets using isotopic evaluation

Brandon W. McDonald[1,2], Troi Perkins[1,2], Robert R. Dunn[3,4], Jennifer McDonald[5], Holly Cole[5], Robert S. Feranec[6] and Roland Kays[1,2]

[1] Department of Forestry and Environmental Resources, North Carolina State University, Raleigh, NC, United States of America
[2] North Carolina Museum of Natural Sciences, Raleigh, NC, United States of America
[3] Department of Applied Ecology, North Carolina State University, Raleigh, NC, United States of America
[4] Natural History Museum of Denmark, University of Copenhagen, Copenhagen, Denmark
[5] Center for Ecology and Conservation, University of Exeter, Penryn, United Kingdom
[6] New York State Museum, Albany, NY, United States of America

Corresponding author
Brandon W. McDonald,
bwmcdona@ncsu.edu

## ABSTRACT

Domestic cats preying on wildlife is a frequent conservation concern but typical approaches for assessing impacts rely on owner reports of prey returned home, which can be biased by inaccurate reporting or by cats consuming prey instead of bringing it home. Isotopes offer an alternative way to quantify broad differences in animal diets. By obtaining samples of pet food from cat owners we predicted that we would have high power to identify cats feeding on wild birds or mammals, given that pet food is thought to have higher C isotope values, due to the pervasive use of corn and/or corn by-products as food ingredients, than native prey. We worked with citizen scientists to quantify the isotopes of 202 cat hair samples and 239 pet food samples from the US and UK. We also characterized the isotopes of 11 likely native prey species from the southeastern US and used mixing models to assess the diet of 47 cats from the same region. Variation in C and N isotope values for cat food was very high, even within the same brand/flavor, suggesting that pet food manufacturers use a wide range of ingredients, and that these may change over time. Cat food and cat hair from the UK had lower C values than the US, presumably reflecting differences in the amount of corn used in the food chains of the two countries. This high variation in pet food reduced our ability to classify cats as hunters of native prey, such that only 43% of the animals could be confidently assigned. If feral or free ranging cats were considered, this uncertainty would be even higher as pet food types would be unknown. Our results question the general assumption that anthropogenic foods always have high C isotope values, because of the high variability we documented within one product type (cat food) and between countries (US vs. UK), and emphasize the need to test a variety of standards before making conclusions from isotope ecology studies.

## INTRODUCTION

While the domestic cat (*Felis catus*) has enjoyed a status as one of the most popular companion animals around the world, they have made a detrimental impact on wildlife (*Medina et al., 2011*). The global population of domestic cats worldwide is estimated at 600 million, including 74 million pet cats and 60–100 million feral cats in the United States (*The Wildlife Society, 2011*; *American Veterinary Medical Association, 2012*; *Jessup, 2004*). Cats can reach very high population densities as a result of artificial food subsidies; thus, even if individual cats have relatively low predation rates because they do not need to hunt in order to eat, the overall effects of cat predation may be severe (*Baker et al., 2005*). Cats have caused intense conservation problems on island systems where there are 33 examples of cats being directly responsible for the extirpation of island species, and another 37 cases in which domestic cats had serious impacts resulting in drastic reductions in populations (*Medina et al., 2011*). This loss or reduction of endemic populations can have alarming impacts on local diversity beyond direct predation pressures. This can vary from reducing the behavioral diversity of potential prey species to disrupting important ecological processes, such as seed dispersal by small vertebrates (*Medina et al., 2014*). The problem continues today, with cats impacting 120 islands throughout the world, resulting in 175 threatened species suffering negative impacts of island cats (*IUCN, 2008*).

The ecological impacts of cats in mainland ecosystems is less clear. Although some estimates suggest that cats kill many billions of birds and mammals each year in the United States (*Loss, Will & Marra, 2013*), the distribution of cats appears to be mostly restricted to areas near development rather than large natural areas, where predators like coyotes (*Canis latrans*) are less likely to occur (*Crooks & Soule, 1999*; *Kays et al., 2015*). Although there are examples of cats affecting prey populations in the mainland (*Loss & Marra, 2017*), there are also examples where no relationship was found (*Kays & Dewan, 2004*; *Sims et al., 2008*; *Lilith, Calver & Garkaklis, 2010*). Nonetheless, there is a general consensus that, due to the high densities of cats in urban areas, they still have a high potential to negatively affect native prey populations (*Woods, McDonald & Harris, 2003*; *Kitts-Morgan, 2015*; *Baker et al., 2008*), and that we should follow a precautionary management strategy (*Calver et al., 2011*).

Most studies into the ecological impact of cats have used owner reports of what they kill, to estimate predation metrics. These studies have revealed great variation in predation rates, ranging from 4–72 prey/cat/year (*Baker et al., 2008*; *Ruxton, Thomas & Wright, 2006*) and are subject to bias as cats tend to only return about 20–30% of prey items caught (*Kays & Dewan, 2004*; *Loyd et al., 2013*). A further complication is that the prey returned to owners does not necessarily reflect what is eaten and how much is eaten. A study in Poland that looked at both owner reports and more direct methods such as fecal analyses and stomach contents revealed that while cats most frequently returned and consumed rodents, mice were brought back much more often than voles, but consumed much less often (*Krauze-Gryz, Gryz & Goszczyński, 2012*). This same study also showed that using pet owners to record prey brought home results in a drastic underestimate to what was being eaten, highlighting the need for methods other than owner reporting to study diet.

Stable isotopes offer an alternative approach to study predator ecology, with $\delta^{13}C$ values reflecting the original source of the carbon as coming from either $C_3$ or $C_4$ plants and $\delta^{15}N$ values typically reflecting trophic level (*Ehleringer et al., 2015*). Because many pet foods use corn (a $C_4$ plant), or corn-fed livestock, as a primary ingredient, cats eating primarily pet food are expected to exhibit much higher $\delta^{13}C$ values than those primarily eating native prey in forested areas, which tend to feed on $C_3$ plants. *Kays & Feranec (2011)* used this approach to classify the range wolves (*Canis lupus*) in the Northeastern U.S. as likely escaped captives or likely natural immigrants, while *Cove et al. (2018)* used it to estimate the percentage of diet for individual feral cats in south Florida. However, both studies used only a few samples of domestic food, which might not reflect the true variety of foods available for pets.

Here we extend the isotopic study of cat diet by focusing on pet cats allowed to spend time inside and outside. By obtaining a sample of the exact food being fed to each cat and comparing these with the isotopic values of their tissue (hair), we expect to have high power to discern the proportion of their diet that comes from pet food and native prey. We also extend sampling of pet food to two countries, the United States (US) and United Kingdom (UK), with different reliance on corn-based food products.

## MATERIALS AND METHODS

### Cat food and hair sample collection

We recruited volunteer pet owners through blog posts, social media, and through a Citizen Science vending machine located at the Oakland Museum of California. Most participants were from Cornwall, UK (106), North Carolina (37), Long Island, NY and southern Connecticut (18), and Oakland, CA (12). The remainder (29) were scattered around the US. Volunteers were asked to send us a small bag of their cat's hair, one tablespoon of their cat's dry pet food and information about how often the cat goes outside, the brand and flavor of the pet food, and notes about any other kinds of food the cat may be consuming (human food, treats, grass, potential prey, etc.). If the cat consumed wet pet food, we purchased a sample of each brand/flavor specified rather than asking volunteers to send a wet sample through the mail. Volunteers were also asked to specify any food changes in the six months prior to collection of hair. We recorded the price of each type of cat food from the US from Chewy.com, except for store specific brands (Harris Teeter$^{TM}$ Your Pet, Trader Joe's$^{TM}$ Cat Food, Kirkland Signature$^{TM}$, and Paws and Claws Delicious Mix$^{TM}$) which were obtained from the websites of those retailers.

Our methods and research are approved by the North Carolina State Institutional Review Board (#3515) and the Animal Care and Use Committee of the North Carolina Museum of Natural Sciences (NCSM 2014-01). All participants filled out an informed consent form prior to providing us with samples and diagnostic information about their cats.

### Sample preparation

All samples of dry food were ground into a uniformly sized powder using a coffee grinder, which was rinsed with water and cleaned with ethyl alcohol between each sample to prevent

contamination. Homogenized samples were stored in separate plastic containers and dried at 40 degrees Celsius for a minimum of 24 h to remove any excess moisture. All wet food was freeze dried and ground into a powder. Small portions of the samples of cat hair provided were treated with a chloroform mixture containing a ratio of 1.0:2.0:0.8 parts chloroform, methanol, and water respectively for one hour in a sonic bath, and then rinsed with deionized water before drying the samples at 40 °C (*Kays & Feranec, 2011*). Samples of hair and feathers of potential prey items were obtained from the Bird and Mammal Collections at the North Carolina Museum of Natural Sciences and were treated using the same process as was used for preparing the samples of cat hair.

Once cleaned and/or dried, all samples (hair, feathers, dry food, and wet food) were weighed using an electric balance to a precision of 0.1 mg into roughly 2 mg portions (we accepted weights between 1.8 and 2.2 mg) and placed into 5 × 9 mm tin capsules. Each tin capsule was folded into a cube and placed into one of three 96 well plastic sample trays. Sample trays from US were sent to the Center for Stable Isotope Biogeochemistry at the University of California at Berkeley for analysis while samples from the UK were sent to Elemtex Stable Isotope Analysis. Samples at each location underwent the same process to determine isotopic composition. Isotopes were reported to a long-term precision of 0.1‰ for $\delta^{13}C$ and 0.2‰ for $\delta^{15}N$.

## Analysis

Cats from the southeastern US were run through a Bayesian two-tracer mixing model using the MixSIAR package in R (*R Core Team, 2013*; *Stock & Semmens, 2016*), which uses a Markov Chain Monte Carlo (MCMC) simulation to estimate the components of each individual cat's diet by modeling the proportions of its sources. Only these cats were considered as all reliable prey isotope values obtained were from this region. To simplify the model, all native prey was averaged into one source and compared against the different types of pet food each cat was known to eat. All models were run with the normal setting of MCMC in the MixSIAR GUI, which has a chain length of 100,000 on three chains. The MixSIAR model also incorporates uncertainty due to the discrimination factor between consumers and sources into the model. Because there are no species-specific discrimination factors calculated for domestic cats to account for changes in isotope values as food sources are incorporated into cat tissue, we created our own value using the diet of one indoor-only cat which ate only one type of cat food. We used the difference in the isotopic value between that cat's food and hair (+1.9‰ for $\delta^{15}N$ and +2.6‰ for $\delta^{13}C$) as a discrimination factor for the rest of our cats, which was lower in adjustment for $\delta^{15}N$ compared to the typical value used by carnivore studies (+3.2‰ for $\delta^{15}N$ and +2.6‰ for $\delta^{13}C$), derived from red fox (*Roth & Hobson, 2000*).

Because the mixing models require all food sources to have n>1 for the complete model to run, we combined foods made by the same manufacturer into brand averages, keeping dry food, fish based wet food, and meat based wet food in separate categories.

Results were compared statistically using Geweke and Gelman–Rubin diagnostics to verify models and then using percent credible intervals (posterior probabilities) as a primary basis of determining relative contributions of different items to the diet of each

cat. Isospace plots and scaled posterior density charts were used as a visual aid to this process. Cats were then categorized into those that likely consumed prey, those that likely ate only cat food, those that had an uncertain diet, and those that had isotope values outside range of food sources. This type of analysis works regionally in areas where the vegetation relies on mostly $C_3$ photosynthesis, and pet food is mostly composed from ingredients that come from $C_4$ plants. This results in native plants and animals having lower $\delta^{13}C$ values than those in pet food.

## Cat classification

Cats were categorized into one of four categories after being run through the MixSIAR mixing model based on the median posterior probabilities generated by the MixSIAR model and isospace plots. The categories were as follows: cats that likely ate some prey, labelled as Hunters; cats that likely eat only cat food, labelled as Non-hunters; cats that had an uncertain diet, labelled as Uncertain; and cats that had high carbon isotope values outside reasonable ranges from food sources, labelled as High-C. Because the mixing models cannot run with a food source represented by a single sample, we used an average of potential prey values (CombinedPrey) and also averaged individual food samples from the same brand. Some brands of food still only had one sample, and thus could be evaluated with isospace plots but not mixing models. Cats were considered to be Hunters if the median proportion value, the 50% credible interval of the posterior probabilities generated through the MixSIAR model, was greater than or equal to 0.245 and the isospace plot showed the cat as being distinguishably closer to the prey source than the cat food source(s). Cats were labelled as Non-hunters if the median proportion value was less than 0.245 with a relatively small amount of variation in the 95% confidence interval generated from the 2.5% and 97.5% credible intervals generated by the MixSIAR model and the isospace plot showed the cat as being distinguishably closer to the cat food source(s). Cats were labelled as High-C when the isospace plot showed the cat as having a more positive carbon value that was outside the range of variation for all potential dietary sources.

Because the MixSIAR model constructs a proposed diet only for the sources input into the model, the posterior probability values were not considered for cats that fit the High-C category, as an unknown source affecting their diet was not included in the model. Cats were labelled as Uncertain when the median proportion value was less than 0.245 but there was a large amount of variation shown by the 95% confidence interval generated by the 2.5% and 97.5% credible intervals and were roughly equidistant from all potential dietary sources on the isospace plot.

# RESULTS

## Isotope values

We obtained isotope values for 47 samples from 13 species of potential prey, while 6 prey samples did not run correctly: 2 samples returned no data and 4 returned extraneous values that did not fit the data (Table S1). Values of $\delta^{13}C$ and $\delta^{15}N$ for all potential native prey from the southeastern US ranged from $-17.7‰$ to $-25.3‰$ and $+2.0‰$ to $+9.5‰$ respectively and averaged $-21.9‰$ ($\pm2.1$ SD) for $\delta^{13}C$ and $+5.5‰$ ($\pm1.7$ SD)
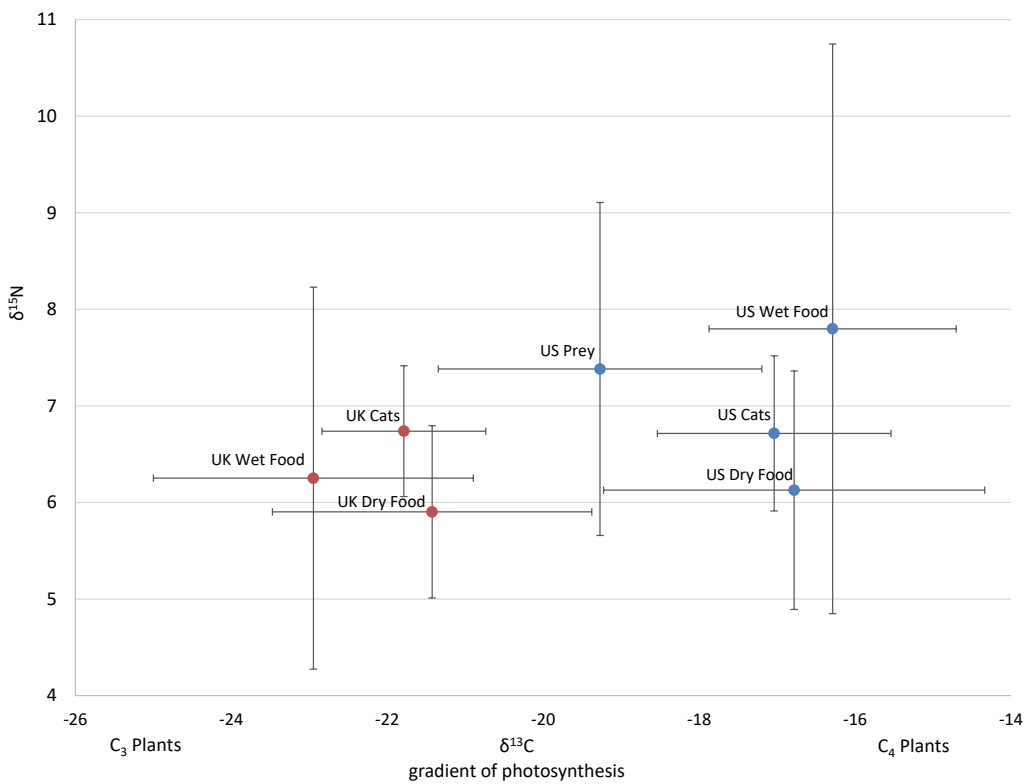

**Figure 1 Average isotope values with standard deviations of cats and food sources from the US and UK** Food and prey values are adjusted by a trophic enrichment factor of $+2.6‰$ $\delta13C$ and $+1.9‰$ $\delta15N$. Carbon values are more negative for both food and cats in the UK. Higher $\delta13C$ values in US cats and food are probably the result of the incorporation of more corn (a C4 photosynthesizing plant) products into cat food, either directly, or through corn-fed livestock. Large variation in $\delta15N$ values for wet food is probably the result of having ocean fish such as tuna, which feed at a higher trophic level and therefore have higher N isotopes.

for $\delta^{15}N$. Individuals of the same prey species tended to have similar overall isotope values. Only 7 prey samples had $\delta^{13}C$ values higher than $-20‰$, including all *Mus musculus* samples, 1 *Pipilo erythrophthalmus*, 1 *Cardinalis cardinalis*, and 1 *Zenaida macroura*.

From US participants we obtained isotope values for 98 samples of dry cat food and 28 samples of wet cat food from 27 different brands representing 55 different flavors of cat food (Table S1). Dry cat food values of $\delta^{13}C$ and $\delta^{15}N$ averaged $-19.3‰$ ($\pm2.3$ SD) and $+4.2‰$ ($\pm1.3$ SD) respectively while wet food values averaged $-18.9‰$ ($\pm1.6$ SD) and $+5.9‰$ ($\pm3.0$ SD) respectively. There was wide variation across cat foods with carbon values ranging nearly $10‰$ and nitrogen values nearly $15‰$ (Fig. 1; Fig. S1). The carbon variation was relatively uniform. Nitrogen variation included one outlier from a tuna-based food, with the next highest value $7‰$ lower. Two-tailed $T$-Tests revealed that $\delta^{13}C$ values differed significantly between potential prey and dry food ($p < 0.001$) as well as wet food ($p < 0.001$). Values for $\delta^{15}N$ differed significantly between potential prey and dry food ($p = 0.007$) but did not differ significantly between wet food and prey.

The 112 food samples from the UK consisted of 61 samples of dry food and 51 samples of wet food. These samples exhibited wide variation with carbon values ranging over 11‰ and nitrogen values ranging nearly 13‰ (Table S2). Food from the UK was also distinctly lower in carbon values compared to food from the US (Fig. 1; Fig. S2). We had multiple samples for 7 US brands, and for 15 US flavors. We found high variation within a brand, with $\delta^{13}$C varying as much as 6.1‰, and $\delta^{15}$N as much as 6.3‰ within a brand (excluding the outlier from tuna food, Table S1, Fig. 2A).

We included isotope values for 96 individual cats from the US of which 47 were female and 49 were male. Cats averaged 8.7 years old ($\pm$3.7 SD), but ranged from 1 to 17 years. Isotope values of $\delta^{13}$C and $\delta^{15}$N for cats averaged -16.9‰ ($\pm$1.6 SD) and +6.8‰ ($\pm$0.9 SD) respectively. Cats also had a fair amount of variation, carbon values ranged about 6.1‰ and nitrogen values ranged about 5.8‰.

Isotope values of $\delta^{13}$C and $\delta^{15}$N for 106 cats from the UK averaged $-21.8$‰ ($\pm$1.1 SD) and +6.7‰ ($\pm$0.7 SD) respectively. Carbon values for cats varied a fair amount ranging about 6‰ while nitrogen values only ranged about 3‰. Age and sex data were not included for the cats from the UK. Two-tailed $T$-tests showed that carbon values differed significantly between the US and UK for cats ($p < 0.001$), dry food ($p < 0.001$), and wet food ($p < 0.001$), while nitrogen values only differed significantly for wet food ($p = 0.017$).

## Classifying cats

Of the 47 cats from the southeastern US that we could compare to regional prey 38 cats were successfully run through the MixSIAR model, while 9 were unable to be run due to having food sources with only one value. All cats successfully run through the model met the criteria of the Geweke and Gelman–Rubin tests. There were 13 cats classified as hunters (Fig. 3A), 7 cats classified as non-hunters (Fig. 3B), and 11 cats classified as high carbon (Fig. 3C). All but one of the 11 cats classified as high carbon would have been classified as hunters based on posterior probabilities in MixSIAR. The remaining 16 cats were unable to be accurately classified based on the mixing models due to intermediate posterior probability values and high levels of variation and were labeled as uncertain (Fig. 3D). Plotting cats by assigned category in an isospace plot of all possible pet food showed extensive overlap between categories (Fig. 4).

Average isotope values for both cats and food differed a bit between categories. Mean cat $\delta^{13}$C values were significantly different in hunters ($-16.99$‰ $\pm$ 1.8 SD) and High-C cats ($-16.55$‰ $\pm$ 0.9 SD) than those in non-hunters ($-15.65$‰ $\pm$ 0.5 SD, $p = 0.02$ and $p = 0.01$ respectively). The mean cat $\delta^{13}$C values of hunters and High-C cats did not differ significantly ($p = 0.44$). There were no significant differences in mean cat $\delta^{15}$N values between all categories; all categories had means between +6.52 and +6.88‰ $\delta^{15}$N. The mean food $\delta^{13}$C value of High-C cats ($-20.57$‰ $\pm$ 1.4 SD) was significantly different than that of hunters ($-18.10$‰ $\pm$ 2.2 SD, $p = 0.003$) and non-hunters ($-19.06$‰ $\pm$ 0.9 SD, $p = 0.014$). There was no significant difference in the mean food $\delta^{13}$C values of hunters and non-hunters ($p = 0.18$). Mean food $\delta^{15}$N values were significantly different ($p = 0.007$) between non-hunters (+5.49‰ $\pm$ 1.1 SD) and High-C cats (+3.88‰ $\pm$ 0.5 SD). The

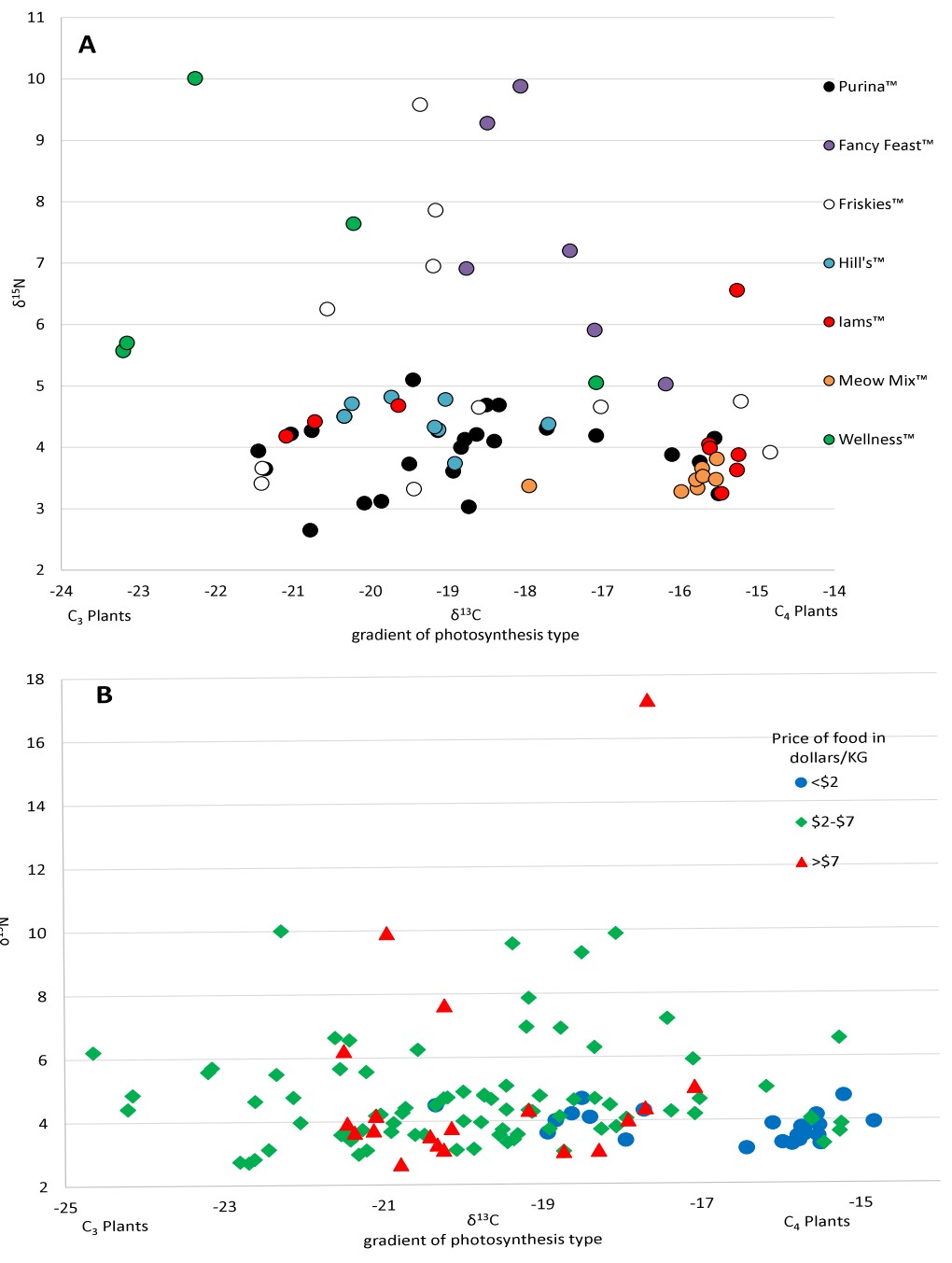

**Figure 2** **Variation in isotopic values of US cat food samples by brand (A) and price (B).** There was a high variance of samples from the same brand, which was partly explained by some of the less expensive foods have higher Carbon isotope values, probably reflecting higher levels of corn, a C4 plant.

mean food $\delta^{15}N$ value for hunters ($+4.45\%_0 \pm 1.2$ SD) did not differ significantly from non-hunters and High-C cats ($p = 0.076$ and $p = 0.155$ respectively).

A two-tailed T-Test revealed sex did not account for any difference in the likelihood of a cat being assigned to a category ($p = 0.63$). Cats ranged between 1 and 17 years old. Age
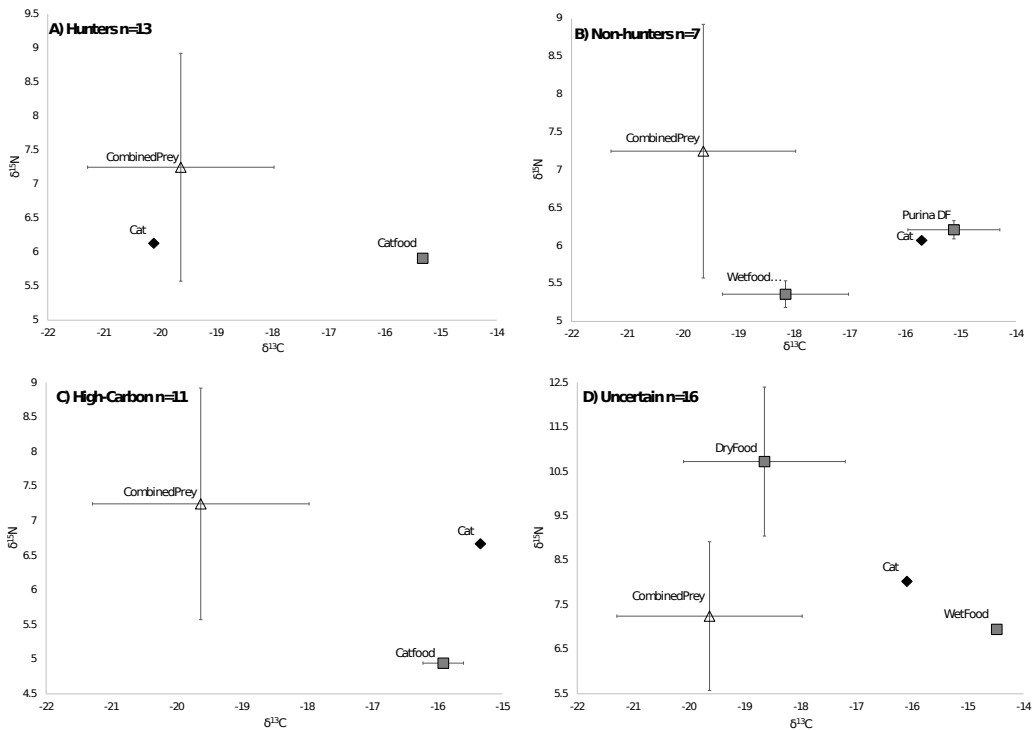

**Figure 3** **Classifications of cats by diet.** Cats were classified by comparing the isotope signatures from their hair to their pet food and potential native prey species. Of the 47 cats included in the analysis, 13 cats were classified as hunters (Fig. 3A), seven cats were classified as non-hunters likely eating mostly pet food (Fig. 3B), 11 cats were classified as having relatively high carbon values compared to potential dietary inputs (Fig. 3C), and 16 cats were classified as having an uncertain diet (Fig. 3D). Characteristic individual isospace plots for each category of cat are shown.

had little impact on the likelihood of cats being assigned to any category; there was a very weak correlation ($R^2 = 0.0093$) between age and category.

Prices for US pet foods ranged from \$1.24 per kilogram to nearly \$25 per kilogram. Food price per unit was significantly negatively correlated with carbon values ($R^2 = 0.108$, $p = 0.00018$), but had no relationship with nitrogen values ($R^2 = 0.0144$, $p = 0.18$) (Fig. 2B). Comparing the average food price per kilogram between the assigned categories of cats using a two tailed $T$-test revealed no significant difference between hunters (\$3.87/kg $\pm$\$2.90 SD) and non-hunters (\$3.16/kg $\pm$\$1.00 SD, $p = 0.43$), and no significant difference between hunters and cats classified as High-C (\$5.44/kg $\pm$ \$2.77 SD, $p = 0.19$). There was a significant difference between non-hunters and High-C cats ($p = 0.03$).

Cats grouped in the uncertain category also had several significant differences from cats in other categories. The mean cat $\delta^{13}$C value of uncertain cats ($-17.83\textperthousand \pm 1.4$ SD) was significantly different from non-hunters ($p < 0.001$) and High-C cats ($p = 0.007$). The mean food $\delta^{13}$C value of uncertain cats ($-19.66\textperthousand \pm 0.9$ SD) was significantly different from hunters ($p = 0.028$). The mean food $\delta^{15}$N value of uncertain cats ($+6.22\textperthousand \pm 1.6$ SD) was significantly different from hunters ($p = 0.002$) and High-C cats ($p < 0.001$). The
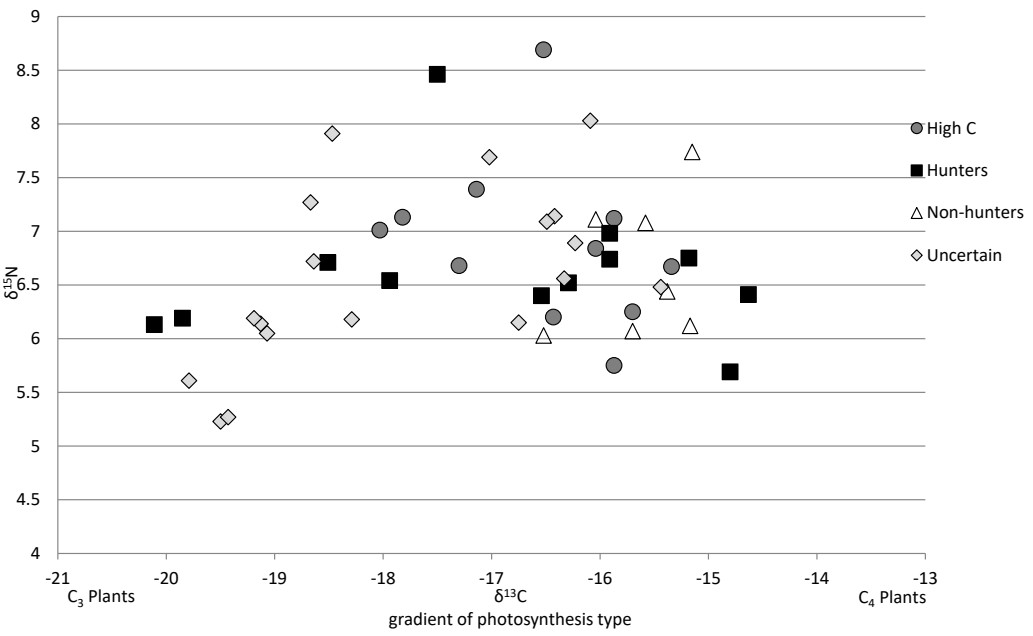

**Figure 4 Isotope values for 47 cats classified by diet category.** All 47 cats represented are from southeastern US as mixing models were derived from cats in this region. By including the pet food, the models were able to detect wild prey in the diet of some individuals, but this plot shows broad overlap of these categories in isospace, due to the high variability of the isotopic values in pet food.

average food price per kilogram of uncertain cats ($4.97/kg ±$1.95 SD) was significantly different from non-hunters ($p = 0.006$).

## DISCUSSION

Because most pet cats are consistently fed one kind of food at home, and because many pet foods are thought to include corn or corn-fed livestock (*Schnepf, 2011*; *Galera et al., 2019*), we expected that a stable isotope approach would have a high power to discern pets consuming native prey from those eating only pet food. However, we found that the high variability of isotope values for pet foods, across and within brands, not to mention across countries, makes it extremely difficult to determine the diet of individual cats. By limiting the potential cat foods for each individual to what their owners fed them, we were able to use mixing models to classify 28% of cats we sampled in the southeastern US as likely hunters, 15% as non-hunters, while 34% were unknown and 23% had unexplainably high C values (Fig. 3). However, if we did not have information from owners about the brand of food they use, for example, as would be the case for feral cats, it would be impossible to categorize the diets of cats with any confidence based on isotope values alone (Fig. 4). It is also important to consider the fact that cats frequently only consume a portion of the prey they kill, potentially leaving only very small influences on the overall diet of an animal. Therefore, counter to our expectations, we found that isotope analysis of cat hair has little value in determining if a cat has been eating wild prey as a result of both high variability in the carbon and nitrogen values of pet foods and the hunting habits of cats.

The high level of variation between different cat food, including many of the flavors produced by the same manufacturers, was quite surprising (Fig. 2A). We received most of our samples of cat food directly from owners and thus only had broad descriptions of the flavors from the volunteers who sent them. Having access to the labels with the nutrition information and ingredients may have helped to understand the variability between foods, however many pet food labels lack specific information about the main sources of carbohydrates, protein, and fat provided and can be misleading (*Galera et al., 2019*). Among dry food alone there was a range of nearly 10‰ in $\delta^{13}$C values. In comparison, all species of our native prey combined had $\delta^{13}$C values ranging 7.6‰. Though $\delta^{13}$C values in both wet food and dry food were significantly different from the values of potential prey, the variation within cat food $\delta^{13}$C values still encompassed a large amount of the variation in the values of potential prey, resulting in difficulty in isolating food sources (see Fig. S1). The largest amount of variation in $\delta^{13}$C values from a single mammalian prey source was 3.54‰ from the woodland vole, while for a single bird prey source it was 6.21‰ from the eastern towhee. It is also important to consider that many of the animals selected as potential prey have diets consisting of a variety of different plants and/or invertebrates. In an animal that eats only one or a select few food sources, such as a grazing Merino sheep, the $\delta^{13}$C value range is 0.84‰ (*Männel, Auerswald & Schnyder, 2007*). The ranges in food isotope values we obtained are very similar to those obtained in a recent Brazilian study looking at the contents of food for domestic dogs (*Galera et al., 2019*). The cats in our study, from both the US and UK, had $\delta^{13}$C value ranges of around 6‰. Cats classified as hunters also had a similar range. Considering that many of these animals are supposedly only consuming pet food, we would expect to see little variation in cats, especially those classified as non-hunters. However, the variation from different types of pet food makes it quite difficult to differentiate dietary inputs. The samples received from participants and the wet food purchased from stores in the Raleigh, NC area made up a total of 27 brands, and 55 individual flavors of food. Purina[TM] was the most frequent brand of cat food received and was made up of 14 different flavors of cat food. Some isotopic variation between flavors would be expected here, but surprisingly, $\delta^{13}$C values varied more than 5‰ across Purina[TM] dry foods. Even foods within the same flavor could be variable, for example, samples of Meow Mix[TM] Original Choice flavor varied 2‰ in $\delta^{13}$C values. This suggests that large scale pet food manufacturers vary the ingredients over time. The only clear relationship with the isotope values we obtained from pet food from the US was that the least expensive foods tended to have higher $\delta^{13}$C values, with most foods priced less than a dollar per kilogram forming a cluster distinct from other foods on an isospace chart at around −16‰ (Fig. 2B). This may be the result of the incorporation of more corn and corn byproducts into inexpensive foods.

Cat foods from the United Kingdom ranged over a similar level of variation as those in the US, but their carbon values were noticeably lower (Fig. S2). The difference in C can likely be explained by the fact that less corn is used in pet foods within the UK (*Howsam, 2018*). Statistics on corn use for animal feed in both countries shows that nearly 700 times as many tonnes of corn is used in the US, and that corn makes up a much larger proportion of the ingredients in animal feed in the US as well (*Schnepf, 2011*). Wheat and barley are

the primary grain ingredients in animal feed in the UK (*Howsam, 2018*), both of which use the C $_3$ photosynthetic pathway, again contributing to the lower $\delta^{13}C$ values in this food. This difference brings into question the extent that high C isotopic values can be used as a global indication of domestic foods in the diet of wild animals (*Penick, Savage & Dunn, 2015*), however, we do not mean to say that using isotope analysis is not both useful and very appropriate in other situations as it has been proven that the use of stable isotopes can aid in answering complex diet and food web problems (*Post, 2002*).

By restricting our comparisons to only the cat food brands eaten by an individual cat, our models classified 13 as hunters, 7 as non-hunters, and 16 as having an uncertain diet. This high level of uncertainty leaves us unable to offer broader generalizations as to the true proportion of cats in our study that consumed native prey. Both hunters and non-hunters were fed less expensive food on average than were cats classified as "uncertain", though the overall average price per kilogram of food only differed significantly between non-hunters and uncertain cats. Since less expensive foods tended to be higher in $\delta^{13}C$ values, these foods had values more different than those of native prey, giving them a greater potential to be categorized as either hunters or non-hunters due to a wider gap between sources. This is probably because the higher $\delta^{13}C$ values of inexpensive foods allowed better discrimination against the low $\delta^{13}C$ values typical of most native prey. However, in the case of hunters, an alternative explanation could be that poor nutrition associated with less expensive food motivates cats to supplement their diet with more native prey. Indeed, cats given a choice of multiple pet foods will select foods to balance their macronutrients (*Hewson-Hughes et al., 2012*). This supports the possibility that cats with access to only one pet food, especially an inexpensive grain-based food, might be more likely to hunt to supplement their diet. This is especially interesting considering that less expensive food is likely what is given to managed feral cat colonies and could potentially be resulting in negative consequences. Testing this hypothesis would require non-isotope diet data to avoid confounding effects of less expensive food offering better isotopic discrimination, due to its typically higher $\delta^{13}C$ values.

The other 11 cats that were categorized as having a high carbon diet had carbon isotope values that were more positive than any potential food sources, both wildlife and cat food. The average $\delta^{13}C$ values of these cats were very similar to those of Hunters, however, the food they were given was more expensive than cats classified as hunters, and significantly more so than non-hunters (Fig. 2B). This more expensive food also tended to have slightly lower $\delta^{13}C$ values making the cats visually appear as though they had higher $\delta^{13}C$ values. It is possible these cats are being supplemented with less expensive cat foods, or potentially table scraps, in addition to what was reported, contributing to $\delta^{13}C$ values higher than that of the food they normally eat.

Cats were ancestrally obligate carnivores. In contrast to dogs, little evidence exists for omnivory among felids in general or the wild relatives of domestic cats. By 7500 BCE cats show some evidence of domestication (*Ottoni et al., 2016*), and by 5300 BCE some populations of cats became partially dependent on food sources associated with human settlements (*Hu et al., 2014*). However, while dogs were able to eat a variety of foods, including some plant matter, early domestic cats appear to have been primarily preying

on other species that were eating grain, rather than grain or other plant matter itself. As a result, while dogs evolved a number of adaptations for feeding on starches during domestication, including multiple copies of amylase genes (*Axelsson et al., 2013*), the same does not appear to be the case for cats (*Di Cerbo et al., 2017*). In this light, it is interesting to consider the modern diets of cats. This diet appears to be different from their ancestral diet in several ways. First, cats now consume different prey species than they might have historically. For example, tuna and other large fish are present in many varieties of cat food. Second, at least some cats have diets in which grains feature prominently. Third, and perhaps most generally, cats now have diets that vary greatly both because of where cats live and because of the fluctuating composition of cat foods. It will be interesting to consider the ways in which this change might be expected to affect the gut microbes, nutrition, health and well-being of cats.

Any isotope ecology study needs to consider the trophic enrichment/discrimination factor, which describes the change in isotope values as food is incorporated into the tissue of an animal. This factor varies across species (*Caut, Angulo & Courchamp, 2009*), and most studies of carnivores have used the factor $+3.2‰$ for $\delta^{15}N$ and $+2.6‰$ for $\delta^{13}C$ derived for red foxes by *Roth & Hobson (2000)*. Preliminary analyses of our data suggested that this adjustment value might not represent cats well, as also noted by other feline studies (*Newsome et al., 2015*; *Parng, Crumpacker & Kurle, 2014*). Therefore, we took advantage of one of our study animals that was a non-hunting indoor only cat fed only one type of dry cat food to derive a cat specific discrimination factor of $+1.9‰$ for $\delta^{15}N$ and $+2.6‰$ for $\delta^{13}C$. This is identical to the C adjustment as found by *Roth & Hobson (2000)* but lower than their N adjustment. Using this value improved the fit of the models in our study, resulting in many more cats appearing in the bounds of reasonable variation between food sources and isospace plots. However, we recognize that this post-hoc analysis of one animal is not a replacement for a controlled experiment with multiple individuals and suggest that future studies should consider adding these to their protocol to improve the discrimination factor estimation for cats. In addition to the need to combine individual food samples into brand averages and the approximated trophic enrichment factor, this study faced several other limitations. A relatively high degree of overlap between the isotope values of potential prey sources and pet food sources limited our ability to identify clear distinctions between the two. Our comparison of native prey was restricted to cats living in the southeastern US, where we had adequate sampling of potential bird and mammal prey, although no values for arthropods or herpetofauna, which occasionally are killed by cats (*Woods, McDonald & Harris, 2003*; *Van Heezik et al., 2010*; *Doherty et al., 2015*). Finally, domestic cats have been proven to not always eat what they kill when hunting. Research shows that often nearly 50% of the kills of outdoor owned cats are left at the site (*Loyd et al., 2013*). This limits the effectiveness of using diet alone to identify the impacts on local wildlife.

## CONCLUSIONS

Stable isotope analysis is a useful tool in identifying broad dietary patterns across trophic levels (from N) and the origin of plant material (from C). However, due to the surprisingly

high level of variation found in isotopic values across different pet foods, we conclude that it is very difficult to distinguish the diet of a cat based on its isotopic values, and thus of little value for future studies of cats predation on native species. Indeed, even though we knew the brand of cat food eaten by an individual cat we were still not able to classify them as a hunter or non-hunter in 57% of the cases; this would be considerably more difficult in the case of feral or free-ranging cats where the exact brand of pet food is unknown or variable. Furthermore, even perfect isotope studies would be unable to account for the prey killed by cats but not eaten. At least 28% of the cats in our study showed evidence of having consumed wild prey, while this is probably an underestimate, it confirms the risk cats can pose to native prey and the importance of studying the phenomenon more. To truly understand the impacts cats have on the wildlife we agree with *Krauze-Gryz, Gryz & Goszczyński (2012)* that a diversity of approaches will be needed since "Not everything is brought home, certain species are eaten more preferentially to others, and small prey may go unnoticed."

## ACKNOWLEDGEMENTS

We would like to thank the volunteers who participated in this, and other Cat Tracker studies, Tim Walsh for assisting with Cat Tracker volunteers through the Bruce Museum in Connecticut, and Lea Shell and Neil McCoy at North Carolina State University for their help in organizing the Cat Tracker email and website through Your Wildlife.

### Funding
This work was supported by the National Science Foundation Student's Discover (No. 1319293), the British Ecological Society (No. LRB16/1013), and the Undergraduate Research Committee at North Carolina State University. The funders had no role in study design, data collection and analysis, decision to publish, or preparation of the manuscript.

### Grant Disclosures
The following grant information was disclosed by the authors:
National Science Foundation Student's Discover: 1319293.
British Ecological Society: LRB16/1013.
Undergraduate Research Committee at North Carolina State University.

### Competing Interests
The authors declare there are no competing interests.

### Author Contributions
- Brandon W. McDonald analyzed the data, performed the experiments, prepared figures and/or tables, authored or reviewed drafts of the paper, and approved the final draft.
- Troi Perkins and Robert R. Dunn conceived and designed the experiments, authored or reviewed drafts of the paper, and approved the final draft.

- Jennifer McDonald and Holly Cole performed the experiments, authored or reviewed drafts of the paper, and approved the final draft.
- Robert S. Feranec and Roland Kays analyzed the data, conceived and designed the experiments, authored or reviewed drafts of the paper, and approved the final draft.

## Animal Ethics

The following information was supplied relating to ethical approvals (i.e., approving body and any reference numbers):

The Animal Care and Use Committee of the North Carolina Museum of Natural Sciences (NCSM 2014-01) provided full approval for this research.

## Ethics

The following information was supplied relating to ethical approvals (i.e., approving body and any reference numbers):

The North Carolina State University Institutional Review Board (No. 3515) granted ethical approval to carry out the study within its facilities.

## Data Availability

The raw data and code are available in the Supplementary Files.

## Supplemental Information

Supplemental information for this article can be found online at http://dx.doi.org/10.7717/peerj.8337#supplemental-information.

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
