# Peer review of "High variability within pet foods prevents the identification of native species in pet cats’ diets using isotopic evaluation"

_PeerJ, doi:10.7717/peerj.8337_

## Round 0.1 · original submission · Major Revisions

Please consider all reviewer comments carefully and submit your revisions. I would spend more time with the available data on cat food contents and nutrition. The expectation for high C4 in cat foods is not quite right, especially for many common wet foods. I also expect some papers like the below will prove of interest.

https://www.ncbi.nlm.nih.gov/pmc/articles/PMC3629282/

·

Basic reporting

The paper is well written and has appropriate contextualization either for the scientific inquiries as for the methods. However there are some corrections to be made.

The delta (δ) symbol should be italicized throughout. See Bond and Hobson (2012) for guidance on recommended isotope ratio terminology.
* Bond AL, Hobson KA (2012). Reporting stable-isotope ratios in ecology: Recommended terminology, guidelines and best practices. Waterbirds 35:324–331. doi: 10.1675/063.035.0213

Pay attention to superscripts and subscripts.

Improve plots, they are not publishable. Letters and symbols could be larger; use black for letters and lines to make it clearer; correct delta symbols and put mass numbers in superscript; and so on.

Line 145: “0.1‰ δ13C and 0.2‰ δ15N” should be “0.1‰ for δ13C and 0.2‰ for δ15N”, with delta symbols in italic and the mass numbers in superscript.

Line 153: “+1.9‰ δ15N and +2.6‰ δ13C” should be “+1.9‰ for δ15N and +2.6‰ for δ13C”, with delta symbols in italic and the mass numbers in superscript.

Line 154: “δ15N” should be with delta symbols in italic and the mass numbers in superscript.

Line 155: “+3.2‰ δ15N and +2.6‰ δ13C” should be “+3.2‰ for δ15N and +2.6‰ for δ13C”, should be with delta symbols in italic and the mass numbers in superscript.

Lines 166 – 168: Revise the subscripts of C3 and C4, and the superscript and italics of δ13C, throughout the text.

Lines 197 - 198: I think the δ13C and δ15N results would be better presented as follows:
Instead of “Values for all potential 198 native prey from the southeastern US ranged from -17.7‰ to -25.3‰ δ13C and 2.0‰ to 9.5‰ δ15N”
I suggest “Values of δ13C and δ15N for all potential 198 native prey from the southeastern US ranged from -17.7‰ to -25.3‰ and 2.0‰ to 9.5‰, respectively.”
This can be done for all δ13C and δ15N results throughout the text.

Line 198: Add standard deviation in the native prey isotopic values averages.

Line 208: “carbon” should be in lowercase.

Lines 295 - 298: It is unclear what you meant here, maybe you could re-write it.

Lines 318 – 319: Unclear.

Experimental design

I have a few questions regarding the MixSIAR model.

Have you considered using the median proportion values from the MixSIAR output for the cats classification and evaluation of their diets, instead of the posterior density plots? (see Galera et al. 2019). Maybe the posterior density plots could be included in the supplemental material to make more room for the isospace plots and MixSIAR output values. Normally the posterior density plots are used as a visual diagnostic of the model, but not the main result of it.

What were the Geweke and Gelman-Rubin diagnostic tests results? (see Galera et al. 2019).

How did you set the Markov Chain Monte Carlo (MCMC) in your models?

Have you classified cats prior to the MixSIAR modelling and then prepared four sets of sources, mixtures and discrimination sheets to run? If yes, what criteria did you use? Or you ran all the data at once and then proceeded with the classification. Explain it better please.

You could add a third tracer to your model, like C:N ratio, to allow a better distinction between cat food and native prey.

Lines 149 – 152: About the discrimination factor, have you considered using an average of all Pet Food cats? Another option would be using a discrimination factor from native felids from the literature.

Lines 247 – 249: The native prey data varied the same as the cat food. More tracers (like C:N ratio) and larger MCMC might be considered to improve the MixSIAR model in this case.

Validity of the findings

The scientific question and the methods chosen are very pertinent. The authors present robust data, but skipped information regarding the handling of it. There are missing important information about the MixSIAR model that would elucidate some of the limitations faced in the analysis.

Reviewer 2 ·

Basic reporting

This manuscript discussed an important topic, the diet content of domestic animals, and should be of interest to readers of the journal as to society in general. Strengths for the study are the knowledge gap identified by the fact that the isotopic composition of cat hairs was not enough to determine the cat diet.
Concerns are that the results related to the research question (the ecological impact of domestic cats preying on wildlife) were not well discussed. And as the authors state, it is already know that it is very difficult to access the wild prey consumption by cats (line 360; 364-366).

The second part of the Title gives a big attention on the fact that the pet foods are sold as a different product of what they are in truth. However, this aspect of result and the consequences of that to cat health was not discussed throughout the text. I suggest including this aspect on the text or changing the title.

In general, the Introduction is appropriately referenced. The structure of the article is conformed to the acceptable format of the Journal.

The axes of the template of Figure 3 (A1, B1, C1, D1) has not a good resolution on page 42.

In some parts of text the English was not clear. I suggest an English revision. I have made a number of suggestions bellow where improvements could be made.
Overall suggestions:
Line 68 – change impact “from” to impact “of”
Line 71 – insert “that” after suggest
Line 93 – change “studying” to “study”
Line 98 – change “out of” to “the”
Line 135 – change “allowing” to “dry”, and remove “to dry”
Line 140/141 – insert a space after the numbers
Line 145, 153, 154 (and elsewhere) – Subscribe the numbers “13” of δ13C and “15” of δ15N
Line 166, 167 – change “C3” to “C3”and “C4” to “C4”
Line 179 – delete “of the area”
Line 296 – change “potentially” to “potential”
Line 326: delete “an”

Experimental design

The statistical analysis on Materials and Methods section are poorly described. There is not sufficient detail about the determination of isotopic composition. The authors should include the equation used to calculate the isotopic ratios of the cat hairs as the cat food.
The author should use a reference to support the statement of lines 93-95. The paper of Ehleringer et al., 2015, for example, talks about how the use of stable isotope technique can identify differences between source and product. (Ehleringer JR, Chesson LA, Valenzuela LO, Tipple BJ, Martinelli LA. 2015. Stable isotopes trace the truth: from adulterated foods to crime scenes. Elements 11(4):259–264).

The study mentioned that the samples were send to different research centers (Center for Stable Isotope Biogeochemistry and Elemtex Stable Isotope Analysis). The analysis performed in each center used the same process to determine the isotopic composition? That information should be mentioned on the text.

The authors have conducted the research with ethical standards in the field.

Validity of the findings

The conclusions were appropriately stated, however, it was poorly linked to the original research question. It should explore the ecological aspects of those results since the author’s main concern regarding cat diets are the impact on wildlife.

---

## Round 0.2 · Minor Revisions

Please further revise your manuscript, taking the reviewers' comments into account and also paying attention to the pdf submitted by Reviewer 1.

·

Basic reporting

The paper is well reported and the figures have been improved. I still have a few comments that can be found in the reviewing PDF.

Experimental design

Material and methods have been improved and made clearer.

Validity of the findings

The idea of the paper is very interesting. Results are compelling and may serve as basis for further studies on the impacts of pet cats on wildlife.

Additional comments

I think the manuscript is almost ready to be published. Take into consideration the comments I've made in the reviewing PDF.

Reviewer 2 ·

Basic reporting

The first sentence of the Introduction section should be appropriately referenced. For example, by the work of Medina (2011) used bellow.
You have made a better connection of the results with the research question, however I suggest that you improve the evidence of domestic cats affecting wildlife at lines 61-69 to provide more justification for your study. For example, what are the consequences of native species extirpation and threaten?

English written:
Lines 64-66: Note that the word “severe” is used three times in a row.
Synonyms for this word could be “alarming”, “critical”, “strong”, “intense”.
The use of the word “island” on lines 66 and 67 seams to be redundant.

Experimental design

Line 216: Please, specify the 112 food samples from UK. Are them dry cat food or wet cat food?

Validity of the findings

Lines 288-290: I suggest reformulating the sentence since the difficulty to determine the impact of domestic cat over native species is not just consequence of the isotope analysis it can be also related to the behavior of the domestic cats that in some cases do not ingest the species they might be hunting - and not eating.

Lines 291-292: Explore the fact that the access to the pet food labels would help to understand the high variability of isotopic carbon composition between different cat foods and the results interpretation, however there is a lack of information about the main sources of protein, carbohydrates, and others on pet food labels (Aro Galera, 2019). Besides the information of ingredients listed on labels can be incomplete.

The information stated on lines 324-326 can be controversial. The main object of your study is the diet of domestic cats and not the influence of human foods on wild animals.
Although in your study you found that the isotopic technic was not enough to answer the main work question (discern the proportion of pet food and native prey source), the composition of carbon isotopes to provide information about the base of a food webs in diets of many animals can be appropriate for others situations (Post, 2002. Ecology v 83, Issue 3 - https://doi.org/10.1890/0012-9658(2002)083[0703:USITET]2.0.CO;2
). Please make that clear in your text.

Additional comments

You have changed the original Tittle but I would suggest a shorter tittle. The words “cat” and “diet” are repeated twice each. You could remove the first line “Isotopic evaluation of cat diet” and leave just “High variability within pet foods prevents the identification of native species in pet cats’ diets using isotopic evaluations”

Note that one of the basic conditions of using stable isotopes on diet studies is that the possible source of food to the animal in study have different isotope value, so that you have a clear answer for the research. Remember to make that clear to the readers.

---

## Round 0.3 · accepted · Accept

Thank you for your careful revisions. I am now happy to accept your paper for publication in PeerJ.